# Association between cigarette smoking and ovarian reserve among women seeking fertility care

Islamiat Oladipupo[1]*, T'shura Ali[1], David W. Hein[2], Kelly Pagidas[3], Henry Bohler[3], Mark A. Doll[2], Merry Lynn Mann[3], Adrienne Gentry[3], Jasmine L. Chiang[3], Rebecca C. Pierson[3], Sashia Torres[1], Emily Reece[1], Kira C. Taylor[1]

1 Department of Epidemiology and Population Health, School of Public Health and Information Sciences, University of Louisville, Louisville, KY, United States of America, 2 Department of Pharmacology and Toxicology, University of Louisville School of Medicine, Louisville, KY, United States of America, 3 Department of Obstetrics, Gynecology, and Women's Health, Division of Reproductive Endocrinology, and Infertility, University of Louisville School of Medicine, Louisville, KY, United States of America

* miyaah82@yahoo.com

**Data Availability Statement:** The data that support the findings of this study are available in Open Science Framework at DOI 10.17605/OSF.IO/NT6P3 under the name "LOUSSI Study".

## Abstract

### Introduction

This study examined the association of smoking with ovarian reserve in a cross-sectional study of 207 women enrolled in the Louisville Tobacco Smoke Exposure, Genetic Susceptibility, and Infertility (LOUSSI) Study and assessed effect modification by NAT2 acetylator phenotype.

### Methods

Information on current smoking status was collected using a structured questionnaire and confirmed by cotinine assay. Serum anti-Müllerian hormone (AMH) levels were used to assess ovarian reserve. Diminished ovarian reserve (DOR) was defined as AMH <1ng/mL. Single nucleotide polymorphisms in the *NAT2* gene, which metabolizes toxins found in cigarette smoke, were analyzed to determine NAT2 acetylator status. Linear and logistic regression were used to determine the effects of smoking on ovarian reserve and evaluate effect modification by NAT2. Regression analyses were stratified by polycystic ovary syndrome (PCOS) status and adjusted for age.

### Results

Current smoking status, either passive or active as measured by urinary cotinine assay, was not significantly associated with DOR. For dose-response assessed using self-report, the odds of DOR increased significantly for every additional cigarette currently smoked (Odds ratio, OR:1.08; 95% confidence interval, 95%CI:1.01–1.15); additionally, every 1 pack-year increase in lifetime exposure was associated with an increased odds of DOR among women without PCOS (OR: 1.08 95%CI: 0.99–1.18). These trends appear to be driven by the heavy or long-term smokers. Effect modification by NAT2 genotype was not established.

**Funding:** a. The LOUSSI study is funded by the National Institutes of Health (NIH), Eunice Kennedy Shriver National Institute of Child Health and Human Development (NICHD)– 1R15HD087911-01 (PI: Kira Taylor). The funders had no role in study design, data collection and analysis, decision to publish, or preparation of the manuscript. https://www.nichd.nih.gov/ b. National Institute of Environmental Health Sciences (NIEHS)– P30ES030283 (PI: Christopher States). The funders had no role in study design, data collection and analysis, decision to publish, or preparation of the manuscript. https://www.niehs.nih.gov/

**Competing interests:** The authors have declared that no competing interests exist.

## Conclusion

A history of heavy smoking may indicate increased risk of diminished ovarian reserve.

## Introduction

Infertility and impaired fecundity among women of reproductive age remain important public health issues. Cigarette smoking in women has been associated with adverse reproductive outcomes such as poorer IVF outcome and increased adverse pregnancy outcomes [1–4]. However, studies of the association of smoking with ovarian reserve have yielded inconsistent results [1, 2, 5–15].

Some studies have reported significantly lower anti-Müllerian hormone (AMH) levels, a measure of ovarian reserve, among smokers [1, 2, 5–9]. Schuh-Huerta et al. however, found that serum AMH levels were significantly higher among smoking women compared to non-smoking women [10]. Other studies found no significant association between smoking and measures of ovarian reserve [11–15]. Previous results are difficult to reconcile as several of the studies [1, 2, 8, 9, 15, 16] were conducted among infertile or sub-fertile populations with none of the studies controlling for polycystic ovary syndrome (PCOS). Women with PCOS diagnosis have been shown to have significantly elevated AMH levels [17, 18].

Genetic heterogeneity is another factor that may explain the inconsistent results of previous studies on the association of smoking with ovarian reserve. N-acetyltransferase2 (NAT2) acetylator status (rapid, intermediate, or slow) has been shown to modify the association of smoking with different disease outcomes, with slow acetylators often more susceptible [19–24]. Taylor et al. reported reduced fecundability among current smokers who are slow acetylators [20]; however, it remains to be established whether these effects on fecundability are modulated through a reduction in ovarian reserve. No studies have assessed the effect of NAT2 acetylator status or the potential interaction with smoking on ovarian reserve.

This study assessed the association of smoking with ovarian reserve in a clinical population of women seeking fertility counseling and used urinary cotinine to validate current smoking status and a smoking questionnaire to estimate cumulative lifetime exposure. In addition, this study explored genetic heterogeneity by assessing whether NAT2 acetylator status influenced the association of smoking with ovarian reserve. Analyses were further stratified by PCOS status to account for higher baseline AMH levels among women with PCOS.

## Materials and methods

### Study population

The Louisville Tobacco Smoke Exposure, Genetic Susceptibility, and Infertility Study (LOUSSI study) is an observational study that is focused on the effects of tobacco exposure and NAT2 acetylator status on ovarian reserve and IVF outcomes. All women 21 years and older seeking infertility treatment in the University of Louisville Reproductive Endocrinology and Infertility (REI) Division were eligible for enrollment into the study. Excluded were women with ongoing pregnancies and patients who could not communicate in English or were unable to understand and complete the informed consent and questionnaire. Women over 45 years of age at the time of initial consult visit were excluded. A total of 261 women were recruited between September 2016 and June 2018. The University of Louisville Institutional Review Board approved this study for human subjects, and all subjects provided written consent prior to participation (IRB number- 16.0063).

## Smoking assessment

Current smoking status was assessed based on urinary cotinine levels. Cotinine, a metabolite of nicotine, has a half-life that ranges from 7 to 40 hours and levels can be used to assess recent exposure to nicotine (past 3–5 days) [25–27]. Urine was collected at enrollment and stored at -80˚C until time of assay. Cotinine ELISA assays (Calbiotech, Spring Valley, CA) were used be estimate urinary cotinine levels with maximum detectable level of cotinine for the assay of 100ng/mL. Urinary cotinine levels were estimated to the nearest 0.50 ng/mL. All self-reported smokers except for one had cotinine levels >100ng/ml; therefore, based on preliminary analysis examining the distribution of cotinine levels in the study population and evidence from literatures [25, 28, 29], the cotinine level used to discriminate smokers from non-smokers was set at ≥14 ng/mL. A "current active smoker" was defined as a person with urinary cotinine levels of ≥14 ng/mL [25]. Women with cotinine levels between 0.5–13.9 ng/mL were classified as "passive smokers" [25]. A "nonsmoker" was defined as a woman with cotinine levels <0.5 ng/mL [25].

Dose-response was assessed using response to a self-administered smoking questionnaire (SSQ). Cumulative lifetime smoking was based on pack-years smoked, calculated for current and former smokers (women with history of smoking at least 1 cigarette/week, who reported quitting more than one month prior to enrollment) by multiplying the number of packs smoked per day by the years of smoking, as reported on the questionnaire.

## NAT2 genotyping

Genomic DNA was isolated from urine samples using the ZR Urine DNA Isolation Kit™ (Zymo Research, Irvine, CA. USA) according to the manufacturer's instructions. NAT2 alleles, haplotypes, genotypes, and deduced phenotype were determined using a NAT2 four-SNP genotype panel of rs1801279 (191G>A), rs1801280 (341T>C), rs1799930 (590G>A) and rs1799931 (857G>A). The accuracy of the four-SNP panel in determining *NAT2* acetylator status is 98.4% and comparable to the seven-SNP panel [30]. The assay uses SNP-specific PCR primers and fluorogenic probes designed using Primer Express™ (Applied Biosystems, CA, USA). The presence of the four SNPs was determined using a predeveloped TaqMan® SNP Genotyping Assay (Applied Biosystems) according to the manufacturer's instructions. Positive and negative (no DNA template) controls were run to ensure that there was no amplification of contaminating DNA, and between 10% and 20% of samples from each plate were run in duplicate. Individuals possessing two of the NAT2 alleles associated with rapid acetylation (*NAT2*4*) were classified as rapid acetylators; individuals possessing one of the alleles associated with rapid acetylation and one allele associated with slow acetylation (*NAT2*5*, *NAT2*6*, *NAT2*7*, and *NAT2*14*) were classified as intermediate acetylators; and those individuals who possessed two slow acetylation alleles (*NAT2*5*, *NAT2*6*, *NAT2*7*, and *NAT2*14*) were classified as slow acetylators.

## Assessment of ovarian reserve

Ovarian reserve can be measured using serum AMH levels or antral follicle count. Serum AMH and antral follicle count (AFC) have strong and similar linear relationship with the size of the primordial follicle pool and ovarian reserve [31–34]. Studies have shown that AMH level and AFC are highly correlated [15, 35]. The limited intra- and inter-cycle variation, objectivity, and potential standardization of AMH assays, makes it the preferred biomarker of ovarian reserve in women.

Assessment of ovarian reserve was based on baseline serum AMH levels extracted from patients' medical records. Women seen in the University of Louisville REI Division routinely

provide a serum sample for analysis of reproductive hormones, including AMH, as part of the initial infertility workup. AMH assays were performed by Quest Diagnostics (Louisville, KY) using chemiluminescence. The limit of detection (LOD) for the assay was 0.03ng/mL; AMH values below the LOD were assigned a value of 0.03ng/mL. For this study, diminished ovarian reserve was defined as baseline serum AMH level less than 1ng/mL [13, 36].

## Statistical analysis

The effect of current active smoking on ovarian reserve was assessed by comparing current active smokers with never smokers and passive smokers. The dose of exposure for current active smoking was based on number of cigarettes smoked per day reported on the SSQ, irrespective of the urinary cotinine values. The dose-response relationship between current active smoking and ovarian reserve was modeled as both a continuous exposure variable and a categorical variable: non-smoker (0 cigarettes/day), moderate smoker (1– <10 cigarettes/day), or heavy smoker (≥ 10 cigarettes/day).

Cumulative lifetime exposure was based on number of packs smoked per day and duration of smoking in years reported on the SSQ by both current active smokers and former smokers. The effect of cumulative lifetime years of active smoking was assessed with both a continuous exposure variable and categorical variable, coded as never smoker, 1–5 pack-years: or >5 pack-years.

Covariates extracted from the medical record included age; race; weight and height at enrollment for calculation of body mass index (BMI); age at menarche; parity; and PCOS status. Age at enrollment was categorized as 21–25, 26–30, 31–35 and 36–45 years. Parity, defined as the number of live births, was put into 3 categories: nulliparous, one live birth, or two or more live births. PCOS status was extracted from the medical record, documented as a current diagnosis or as a previous diagnosis self-reported by the patient. BMI at enrollment was categorized as normal weight (17.5–24.9), overweight (25.0–29.9), obese (30.0–34.9), or morbidly obese (≥35). Differences in demographics and covariates between smokers and non-smokers were tested using a Chi-square/Fisher's exact test for categorical variables, Student's t-test for continuously normally distributed variables, and the Wilcoxon rank-sum test for continuous variables that were not normally distributed.

Linear regression models using the natural logarithm (ln) of AMH as the dependent variable were generated to estimate the effects of the different smoking variables on ovarian reserve, adjusted for covariates. Percent differences and 95% confidence intervals (CIs) in serum AMH between exposure groups were calculated from the parameter estimates as follows: ([exp(β) −1] *100) and presented with corresponding 95% confidence intervals (CIs) and p-values [13]. Unconditional logistic regression models were used to estimate odds ratios (ORs) and 95% CIs for the association of smoking on DOR. Regression analyses were adjusted for variables that were significantly associated with ovarian reserve (age and PCOS).

Effect modification by *NAT2* genotype was assessed by including an interaction term between NAT2 and the smoking variables in the multivariable models. The rapid acetylator phenotype was combined with the intermediate acetylator phenotype to form the referent group for comparison with the slow acetylator group which has been used in other studies [30, 37–39]. Interaction models were further adjusted for race, in addition to other covariates. Ethnic differences in the frequency of rapid and slow acetylator NAT2 alleles or haplotype have been reported [20, 40–42]. A P-value <0.05 was considered statistically significant. All regression models were also run restricted to women without PCOS. All analyses were two-sided and conducted using Statistical Analysis Software version 9.4 (SAS Institute, Cary, NC).

## Results

### Population

Out of 261 women who were enrolled into the LOUSSI study, 54 were excluded due to incomplete records (52 were missing serum AMH levels and 2 were missing PCOS status) (Fig 1). Thus, 207 women were included in the final analysis of the questionnaire data (self-reported dose data such as pack-years). For the analysis of current smoking status using cotinine measurement, 8 of the 207 were excluded as they did not provide urine for cotinine assays (Fig 1).

### Descriptive statistics and bivariable analyses

Of the remaining 199 participants, 29% were current active smokers with 55% of these reporting smoking 10 or more cigarettes per day, 21% smoking between 5 to 9 cigarettes per day and 24% reported smoking fewer than 5 cigarettes per day. Current active smokers were more likely to be of self-reported Black race (p<0.001), obese (p = 0.03) and single (67% vs. 22%,

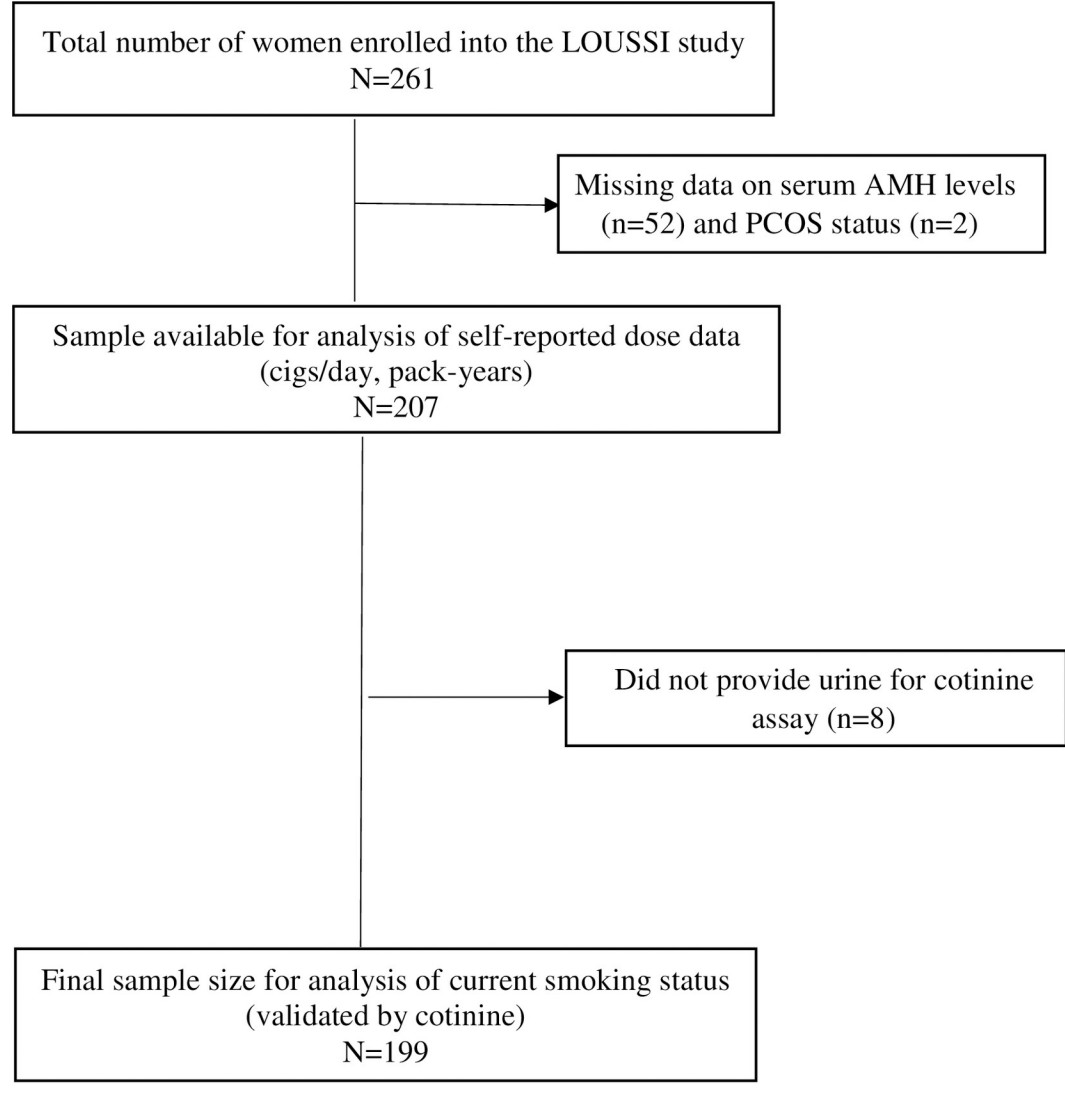

**Fig 1. Population flow diagram.**

p = <0.001), compared with non-smokers (Table 1). There were no significant differences between current active smokers and non-smokers for the other characteristics examined (Table 1).

We also examined the association of each potential covariate with ovarian reserve, as measured by either DOR or % change in AMH. Only age and PCOS status were significantly associated with ovarian reserve in bivariable analyses and were considered as potential confounders in the multivariable analyses. A 48% decrease (95%CI: -71.0 to -8.5) in AMH level was reported among women between ages 31 to 35 years and a 67% decrease (95%CI: -81.0 to -40.6) among women older than 35 years of age, relative to women less than 31 years old. A similar inverse association was found between age and DOR, with women between the ages of 36 to 45 years having 6 times the odds of DOR (OR = 6.2; 95% CI:1.7–22.3) compared to women 21 to 25 years of age. A PCOS diagnosis was associated with a significant increase in AMH levels (177.0%; 95% CI: 89.0–306.1) and a decrease in the odds of DOR (OR = 0.3; 95% CI: 0.1–0.7). Neither serum AMH levels nor DOR was found to be significantly associated with race, BMI, age at menarche, or parity.

## Multivariable modeling results

Current active smoking, based on cotinine measurement, was not significantly associated with DOR (OR = 0.62; 95% CI: 0.24–1.56); neither was passive smoking: (OR = 1.32; 95% CI: 0.59–3.0) (Table 2). Results were similar among women without PCOS. The results were similar when assessment of current smoking was based only on self-report (OR = 1.22; 95% CI: 0.45–3.27; Non PCOS: OR = 1.87; 95% CI: 0.56–6.33).

However, in the analysis of current cigarettes reported per day on the questionnaire, which was analyzed regardless of cotinine levels, current cigarette smoking was associated with an increased odds of diminished ovarian reserve among women without PCOS. On average, the odds of DOR increased by 8% for every additional cigarette currently smoked (OR = 1.08; 95% CI:1.01–1.15). There was also suggestion of a dose-response with self-reported lifetime exposure. For every 1 pack-year increase in lifetime exposure, the odds of DOR increased by 8% (OR = 1.08; 95% CI: 0.99–1.18). Results were similar but not statistically significant when analyses were limited to women with validated cotinine levels and without PCOS (Cigs/day: OR = 1.06; 95% CI: 0.99–1.14; pack-year: OR = 1.08; 95% CI: 0.99–1.18).

We also examined cigarettes per day and pack-years of exposure in categories (Table 2). In these models, the effects of smoking on DOR were only observed in the highest exposure groups (at least 10 cigarettes/day or more than 5 pack-years of exposure), suggesting that the observed associations may be driven by those with high exposure levels.

Adjusted linear regression models for the association of current active smoking and percentage change in AMH are presented in Table 3. There was no significant association found between current smoking, the number of cigarettes smoked, or cumulative lifetime exposure and AMH levels.

## NAT2 interaction analysis

A total of 11 *NAT2* genotypes were identified. The concordance of genotype duplicates was 100% for all four SNPs. DNA extraction or *NAT2* genotyping was unsuccessful in 14 participants, yielding a sample size of 155 for the *NAT2* interaction analysis. Approximately 39% of women in the study were slow acetylators, 44% were intermediate, and 17% were rapid acetylators. The distribution of *NAT2* acetylator status did not differ by smoking status (Table 1) or self-reported race (S3 Table in S1 File).

After adjusting for age, race and PCOS status, there was no statistically significant interaction with NAT2 phenotype (Table 4). Although there was a suggestion that the odds of DOR

**Table 1. Characteristics of women in the LOUSSI study, stratified by smoking status (N = 199)[a].**

| | Nonsmokers (Cotinine < 0.5 ng/mL) | Passive smokers (Cotinine 0.5–13.9 ng/mL) | Active smokers (Cotinine≥14 ng/mL) | P-value |
|---|---|---|---|---|
| | N (%) (n = 71) | N (%) (n = 70) | N (%) (n = 58) | |
| **Age (years)** | | | | 0.06 |
| 21–25 | 6 (8.4) | 10 (14.2) | 15 (25.9) | |
| 26–30 | 13 (18.3) | 13 (18.6) | 14 (24.1) | |
| 31–35 | 23 (32.4) | 27 (38.6) | 12 (20.7) | |
| 36–45 | 29 (40.9) | 20 (28.6) | 17 (29.3) | |
| **Race** | | | | <0.001 |
| White | 38 (53.5) | 37 (53.6) | 29 (50.0) | |
| Black | 12 (16.9) | 20 (29.0) | 27 (46.6) | |
| Other | 21 (29.6) | 12 (17.4) | 2 (3.4) | |
| Missing | | 1 (1.4) | | |
| **Age at menarche(years) mean (SD)[b]** | 12.7 (2.0) | 12.4 (1.7) | 12.2 (1.5) | 0.33 |
| Missing | 2 (2.8) | 4 (5.7) | 1 (1.7) | |
| **Body Mass Index (kg/m$^2$)** | | | | 0.03 |
| 17.5–24.9 | 22 (31.4) | 15 (21.8) | 14 (24.1) | |
| 25–29.9 | 26 (37.1) | 19 (27.5) | 12 (20.7) | |
| 30–34.9 | 13 (18.6) | 12 (17.4) | 18 (31.1) | |
| ≥ 35 | 9 (12.9) | 23 (33.3) | 14 (24.1) | |
| Missing | 1 (1.4) | 1 (1.4) | | |
| **Polycystic Ovary Syndrome** | | | | 0.24 |
| No | 47 (68.1) | 38 (54.3) | 36 (62.1) | |
| Yes | 2 (31.9) | 32 (45.7) | 22 (37.9) | |
| Missing | 2 (2.8) | | | |
| **Parity** | | | | 0.11 |
| None | 26 (36.6) | 35 (51.5) | 16 (28.1) | |
| One | 17 (23.9) | 14 (20.6) | 17 (29.8) | |
| Two or more | 28 (39.5) | 19 (27.9) | 24 (42.1) | |
| Missing | | 2 (2.9) | 1 (1.7) | |
| **Marital Status** | | | | <0.001 |
| Single | 15 (21.7) | 17 (26.6) | 37 (67.3) | |
| Married | 54 (78.3) | 47 (73.4) | 18 (32.7) | |
| Missing | 2 (2.8) | 6 (8.6) | 3 (5.2) | |
| **Serum AMH categories (ng/mL)** | | | | 0.32 |
| <1 | 19 (26.8) | 19 (27.1) | 9 (15.5) | |
| 1–2 | 14 (19.7) | 10 (14.3) | 14 (24.1) | |
| 2–3 | 20 (28.1) | 18 (25.7) | 22 (38.0) | |
| >3 | 18 (25.4) | 23 (32.9) | 13 (22.4) | |
| **Diminished ovarian reserve (DOR) (AMH<1 ng/mL)** | | | | 0.23 |
| No | 52 (73.2) | 51 (72.9) | 49 (84.5) | |
| Yes | 19 (26.8) | 19 (27.1) | 9 (15.5) | |
| **NAT2 Acetylator Status** | | | | |
| Rapid/ Intermediate Acetylators | 40 (61.5) | 39 (61.9) | 18 (62.1) | 0.99 |
| Slow Acetylators | 25 (38.5) | 24 (38.1) | 11 (37.9) | |
| Missing | 6 (8.5) | 7 (10) | 1 (3.3) | |

[a] Women with cotinine measurements.

[b] SD = Standard deviation

**Table 2. Multivariable model for the association of cigarette smoking with diminished ovarian reserve (DOR).**

| | All women[a] | | Women without PCOS[b] | |
|---|---|---|---|---|
| | N | OR (95% CI) for DOR | N | OR (95% CI) for DOR |
| **Current smoking status, using cotinine measurement** | | | | |
| Nonsmoker | 71 | Referent | 47 | Referent |
| Passive smoker | 70 | 1.32 (0.59–3.0) | 38 | 0.85 (0.33–2.19) |
| Active smoker | 58 | 0.62 (0.24–1.56) | 36 | 0.52 (0.18–1.52) |
| **Self-reported cigarettes/day** | | | | |
| Treated continuously | 207 | 1.05 (0.997–1.11) | 125 | 1.08 (1.01–1.15)[c] |
| Categories: | | | | |
| None | 174 | Referent | 107 | Referent |
| 1–9 | 15 | 0.27 (0.03–2.27) | 8 | 0.44 (0.05–4.27) |
| ≥10 | 18 | 2.64 (0.81–8.64) | 10 | 4.58 (0.91–23.02) |
| **Self-reported cumulative lifetime exposure in pack-years** | | | | |
| Treated continuously | 207 | 1.05 (0.97–1.13) | 125 | 1.08 (0.99–1.18) |
| Categories: | | | | |
| Never | 150 | Referent | 93 | Referent |
| 1–5 | 35 | 0.67 (0.24–1.82) | 21 | 1.09 (0.36–3.34) |
| >5 | 22 | 1.48 (0.49–4.47) | 11 | 2.51 (0.61–10.23) |

[a] Models adjusted for age and polycystic ovary syndrome

[b] Models adjusted for age

[c] P-value = 0.04

**Table 3. Multivariable model for the association of cigarette smoking with AMH levels.**

| | All women[a] | | Women without PCOS[b] | |
|---|---|---|---|---|
| | N | % Change in AMH (95% CI) | N | % Change in AMH (95% CI) |
| **Current smoking status, using cotinine measurement** | | | | |
| Nonsmoker | 71 | Referent | 47 | Referent |
| Passive smoker | 70 | -14.2 (-45.3–34.4) | 38 | 1.5 (-44.1–84.4) |
| Active smoker | 58 | 5.2 (-34.5–68.9) | 36 | 60.1 (-14.0–198.3) |
| **Self-reported cigarettes/day** | | | | |
| Treated continuously | 207 | -2.3 (-5.2–0.7) | 125 | -1.3 (-5.1–2.7) |
| Categories: | | | | |
| None | 174 | Referent | 107 | Referent |
| 1–9 | 15 | 9.9 (-45.7–122.1) | 8 | -9.5 (-67.6–152.6) |
| ≥10 | 18 | -38.6 (-67.9–17.4) | 10 | -17.0 (-67.3–110.8) |
| **Self-reported cumulative lifetime exposure in pack-years** | | | | |
| Treated continuously | 207 | -1.5 (-5.70–3.0) | 125 | -1.2 (-6.4–4.3) |
| Categories: | | | | |
| Never | 150 | Referent | 93 | Referent |
| 1–5 | 35 | 9.6 (-33.3–80.2) | 21 | -12.8 (-56.0–72.7) |
| >5 | 22 | -27.6 (-61.4–35.6) | 11 | -7.2 (-63.5–136.1) |

[a] Models adjusted for age and polycystic ovary syndrome

[b] Models adjusted for age

**Table 4. Interaction of current cigarette smoking and *NAT2* acetylator phenotype on diminished ovarian reserve (DOR).**

| | All women[a] | | | | Women without PCOS[b] | | | |
|---|---|---|---|---|---|---|---|---|
| | Rapid/Intermediate | | Slow | | Rapid/Intermediate | | Slow | |
| | **N** | **OR (95% CI)** | **N** | **OR (95% CI)** | **N** | **OR (95% CI)** | **N** | **OR (95% CI)** |
| Nonsmoker | 38 | 1.00 | 25 | 0.42(0.05–3.44) | 26 | 1.00 | 16 | 0.35(0.02–5.14) |
| Passive smoker | 39 | 1.16(0.30–4.50) | 24 | 2.42(0.37–15.97) | 25 | 1.43(0.31–6.66) | 9 | 1.92(0.11–33.78) |
| Active smoker | 18 | 1.25(0.16–9.54) | 11 | 1.59(0.26–9.70) | 13 | 1.71(0.17–17.12) | 3 | 2.50(0.15–42.6) |

[a] Adjusted for age, race, and polycystic ovary syndrome; p-value for interaction = 0.67

[b] Adjusted for age and race; p-value for interaction = 0.69

was higher among slow acetylators (OR = 1.59; 95% CI: 0.26–9.70), the sample size was insufficient to provide adequate statistical power.

## Discussion

In this study of women seeking fertility counseling, current smoking was not associated with diminished ovarian reserve. However, other results and trends observed in this study were consistent with the hypothesis that heavy smoking, or smoking for a long duration, may decrease ovarian reserve. Although most results were not statistically significant at the $\alpha = 0.05$ level, one statistically significant association was observed, between cigarettes smoked per day and odds of diminished ovarian reserve among women without PCOS. We also identified other trends consistent with the hypothesis that smoking reduces ovarian reserve. While the 95% confidence intervals crossed the null value for the other analyses, the magnitude and direction of effect suggested that heavy smoking (>10 cigarettes per day or more than 5 pack-years over the lifespan) reduces ovarian reserve. The effect sizes among women without PCOS were larger, perhaps because the AMH levels among these women are a more accurate reflection of ovarian reserve. Although it is possible that variation in AMH levels among women with PCOS may also reflect ovarian reserve, women with and without PCOS represent two qualitatively different populations with regard to AMH levels.

The validity of this study is supported by its use of a targeted smoking questionnaire, which assessed both current and lifetime smoking habits, and the use of a urinary cotinine assay to validate current smoking. In addition, both diminished ovarian reserve and AMH levels were assessed as outcomes, and results using either outcome were consistent with a dose-response effect of cigarette smoking (current and lifetime) with decreased ovarian reserve. Current active smoking, assessed using cotinine levels, was not significantly associated with DOR. However, there is an established relationship between smoking and earlier age at menopause [43–47], which further increases the plausibility that smoking is associated with decreased ovarian reserve, possibly through accelerated ovarian aging and follicle atresia.

Constituents of tobacco smoke known to have toxic effects on reproductive health include carbon monoxide, nicotine, cadmium, and polycyclic aromatic hydrocarbons (PAHs) such as benzo[a]pyrene (B[a]P) [48, 49]. Some of the proposed biological mechanisms through which tobacco smoke influences ovarian reserve include inhibition of follicular development; premature luteinization of the preovulatory follicle; reduction of oocyte vascularization and maturation; atresia of oocytes in primordial and small primary follicles; impaired steroidogenesis; increased chromosomal errors; and cytotoxicity [49–52].

Several previous studies have suggested increased risk for diminished ovarian reserve with smoking [1, 2, 5–10], but results have been inconsistent. Recent studies did not find a significant association between smoking and biomarkers of ovarian reserve [53–55]. These studies

were limited by small sample size and did not control for PCOS status. However, in a cross-sectional study of 913 premenopausal women, White et al. (2016) found a significant association between self-reported smoking and serum AMH among women who smoked 20 or more cigarettes/day (−56.2%, 95%CI: −80.3, −2.8%) [5]. Similar findings with number of pack-years smoked were reported by Dolleman et al., where a significant association between smoking and age-specific serum AMH was found for 10–15 pack-years (β = -7.0, p = 0.003) and 15–20 pack-years (β = -8.5, p = 0.001) pack-years but not for 5–10 pack-years (β = -3.1, p = 0.15) [6]. Therefore, like the present study, prior studies also found that a clinically meaningful reduction in ovarian reserve was only detected among heavy or long-term smokers.

While it is true that women and their partners trying to conceive have been counseled to quit smoking for decades, smoking was still highly prevalent (58/199, or almost 29%) among these women who were actively attempting to conceive and seeking treatment for fertility counseling. Other studies that reported significant inverse associations between smoking and ovarian reserve (as measured by AMH) also reported moderate [5, 7, 10] to high [1, 2, 8, 9] prevalence of smoking in the study population, emphasizing the continued importance of research in the area of smoking and fertility.

Studies on the association of smoking with ovarian reserve varied in the covariates selected for adjustment. None of the other studies controlled for PCOS, even though several of the studies [1, 2, 8, 9, 16] were conducted among infertile or sub-fertile populations. Women with PCOS diagnosis have been shown to have significantly elevated AMH levels [17, 18]. In analyses that excluded women with PCOS diagnosis, effect sizes were generally stronger, suggesting PCOS is a confounder and/or an effect modifier of the association of smoking with ovarian reserve. Therefore, controlling for and stratifying on PCOS was a major improvement in the study design compared to prior studies.

This study further explored population differences in NAT2 genotype frequencies as a possible reason for the inconsistency in the association of smoking with ovarian reserve. Results from this study, though not statistically significant, suggest that the risk of diminished ovarian reserve may be increased for current smokers who are slow NAT2 acetylators. Polymorphisms in the NAT2 gene modify susceptibility to harmful heterocyclic and aromatic amines constituents in tobacco smoke [40, 56, 57]. High levels of DNA adduct of 4-aminobiphenyl, a carcinogenic aromatic amine in tobacco smoke, have been reported in tissues of smokers such as larynx [58], liver [59], bladder [60, 61], breast [62–64] and sputum [65]. Additionally, it has been shown that slow NAT2 acetylators have variable reductions in catalytic activities that make them more susceptible to the effect of some toxins [40, 56, 57, 66]. Given the significantly reduced fecundability reported among current smokers who are slow acetylators by Taylor et al. [20], it is plausible that the effect is modulated through a reduction in ovarian reserve. Slow NAT2 acetylator phenotypes result from different mechanisms based on the presence of single nucleotide polymorphisms [66, 67], and subsequent studies have shown that the slow phenotype is not homogenous [68, 69]. Therefore, treating all slow acetylators as a single group may be diluting any effects present in a particular haplotype. Genetic heterogeneity of the slow NAT2 phenotype was investigated by assessing the interaction of smoking with individual NAT2 alleles and genotypes on ovarian reserve, but this analysis was limited by small sample size. More studies, with a large sample size, are needed to clarify the role of NAT2 acetylator phenotype as well as specific NAT2 genotypes in the association of smoking with ovarian reserve.

This study has a number of limitations common to cross sectional studies, including the potential for recall bias (e.g., pack-years smoked), resulting in possible misclassification of exposure; potential selection bias; and residual confounding. Recruiting participants from a single clinic may result in lack of generalizability, though patients who seek care at this site

have varied payor mix. In addition, women excluded from this analysis for missing information on important variables (e.g., AMH) may be different in meaningful ways from women who had complete data. Finally, we presented a parsimonious model, and there could be residual confounding. As the associations of socioeconomic status and other covariates such as alcohol and prior use of hormonal contraceptive on ovarian reserve have not been well-established, it is uncertain how these unmeasured covariates may have influenced the effect estimates obtained from this study. Previous diagnosis of endometriosis or a history of ovarian surgery that may diminish ovarian reserve were not assessed in this study.

This study was limited by the small sample of heavy smokers in the study population which reduced the power to detect the main effect of smoking or its potential interaction with NAT2 on ovarian reserve. Consequently, it cannot be certain whether the null associations are attributable to an actual lack of association or to inadequate statistical power. Future research with a similar study design should plan to expand recruitment to multiple gynecological and fertility clinics and should conduct power calculations based on the expected number of heavy smokers.

Strengths of this study include the novel investigation of the impact of NAT2 acetylator status on the relationship of smoking with ovarian reserve; the use of questionnaires that permitted a more detailed characterization of both current and lifetime exposure than typical patient intake questionnaires; and validation of current exposure status with cotinine.

## Conclusions

Current smoking was not associated with diminished ovarian reserve. However, the trends and directions of effect observed in this study suggest that heavy smoking, or smoking for a long duration, may reduce ovarian reserve. Additional data are needed to better define the role of NAT2 polymorphisms with diminished ovarian reserve following exposure to toxins such as those found in cigarette smoke.

## Supporting information

**S1 File.**
(DOCX)

## Acknowledgments

The authors express their sincere thanks to the LOUSSI study participants who made this work possible.

## Author Contributions

**Conceptualization:** Islamiat Oladipupo, David W. Hein, Kira C. Taylor.

**Data curation:** Islamiat Oladipupo, T'shura Ali, Sashia Torres, Emily Reece, Kira C. Taylor.

**Formal analysis:** Islamiat Oladipupo, Kira C. Taylor.

**Funding acquisition:** David W. Hein, Henry Bohler, Kira C. Taylor.

**Investigation:** Islamiat Oladipupo, T'shura Ali, Mark A. Doll, Rebecca C. Pierson, Sashia Torres, Emily Reece, Kira C. Taylor.

**Methodology:** Islamiat Oladipupo, Kira C. Taylor.

**Project administration:** Islamiat Oladipupo, Kira C. Taylor.

**Resources:** David W. Hein, Kelly Pagidas, Henry Bohler, Mark A. Doll, Merry Lynn Mann, Adrienne Gentry, Jasmine L. Chiang, Rebecca C. Pierson, Kira C. Taylor.

**Supervision:** David W. Hein, Kira C. Taylor.

**Visualization:** Islamiat Oladipupo, Kira C. Taylor.

**Writing – original draft:** Islamiat Oladipupo.

**Writing – review & editing:** T'shura Ali, David W. Hein, Kelly Pagidas, Henry Bohler, Mark A. Doll, Merry Lynn Mann, Adrienne Gentry, Jasmine L. Chiang, Rebecca C. Pierson, Sashia Torres, Emily Reece, Kira C. Taylor.

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
