## [Decision Letter · Decision Letter 0]

12 Jul 2022

PONE-D-22-13340ASSOCIATION BETWEEN CIGARETTE SMOKING AND OVARIAN RESERVE AMONG WOMEN SEEKING FERTILITY CAREPLOS ONE

Dear Dr. Oladipupo,

Thank you for submitting your manuscript to PLOS ONE. After careful consideration, we feel that it has merit but does not fully meet PLOS ONE’s publication criteria as it currently stands. Therefore, we invite you to submit a revised version of the manuscript that addresses the points raised during the review process. Please submit your revised manuscript by Aug 12 2022 11:59PM. If you will need more time than this to complete your revisions, please reply to this message or contact the journal office at plosone@plos.org. Please include the following items when submitting your revised manuscript:A rebuttal letter that responds to each point raised by the academic editor and reviewer(s). You should upload this letter as a separate file labeled 'Response to Reviewers'.A marked-up copy of your manuscript that highlights changes made to the original version. You should upload this as a separate file labeled 'Revised Manuscript with Track Changes'.An unmarked version of your revised paper without tracked changes. You should upload this as a separate file labeled 'Manuscript'.

We look forward to receiving your revised manuscript.

Kind regards,

Jing Zhang

Academic Editor

PLOS ONE

Journal Requirements:

**Comments to the Author**

1. Is the manuscript technically sound, and do the data support the conclusions?

Reviewer #1: Partly

Reviewer #2: Yes

2. Has the statistical analysis been performed appropriately and rigorously? 

Reviewer #1: Yes

Reviewer #2: Yes

3. Have the authors made all data underlying the findings in their manuscript fully available?

Reviewer #1: Yes

Reviewer #2: Yes

4. Is the manuscript presented in an intelligible fashion and written in standard English?

Reviewer #1: Yes

Reviewer #2: Yes

5. Review Comments to the Author

Reviewer #1: This paper is of importance for women considering fertility.

Smoking and other addictive substances affect many patients. Warning of nicotine exposure is a continuous challenge.

The importance of segregating patients with PCO who have high AMH enable to better assess the non PCO group.

However lack of definite information on hormonal contraceptives of any kind that can reduce AMH while in use weaken the paper. With BCP or implant a 25% decrease was noted. This has to be examined before conclusion of the results are made. Possibly this is a major contributor for borderline results obtained. Also AMH level that is consistent with PCO needs to be better defined in the methods section based on currently accepted criteria.

Showing the results in a graph format. Where individual results showed provide a better description of data

Reviewer #2: This is a very interesting article based on the results of a cross sectional study conducted with 207 women enrolled in the LOUSSI Study (Louisville Tobacco Smoke Exposure Genetic Susceptibility and Infertility) that fulfils all the inclusion criteria to be accepted for review in this Journal

I believe that the study is important, however I would like the authors to clarify some points:

1) The first sentence of the Results in the Abstract section (lines 28-29 Page 2) should be more clear because seems contradictory with the next one. Maybe authors could mention "Current smoking status either passive or active measured by a urinary cotinine assay..."

2) Although the previous history of pelvic surgery and of endometriosis has been considered the study has the drawback that has been conducted in a population seeking infertility counselling or treatment meaning that are women with infertility problems some of them could be related to diminished ovarian reserve not related to tobacco use. I wonder if authors have information of the same studies conducted in general population without fertility problems

3) It is very interesting the NAT2 acetylation status screening performed and it is a pity that due to the reduced sample size no differences have been observed in relation to the slow, intermediate or rapid categories

4) I am surprised that 37,9 % of never smokers presented cotinine levels > 14 ng/ml. I would like authors to further comment on this finding. Maybe they should be more critical on the validity of this cotinine assay because it proved unable to discriminate what the clinical questionnaire on tobacco exposure did.

5) Tables are busy and perhaps some of them can be simplified

6. PLOS authors have the option to publish the peer review history of their article (what does this mean?). If published, this will include your full peer review and any attached files.

Reviewer #1: **Yes: **Eytan Barnea

Reviewer #2: No

---

## [Author Response · Author response to Decision Letter 0]

1 Sep 2022

Response to Reviewers.

Journal Editor- Journal requirements

1. The manuscript has been updated to meet the PLOS ONE's style requirements, including those for file naming. 

2. For sections where the phrase “data not shown” was used in the manuscript, data has been provided in the supplemental file and phrase removed accordingly. See line 241 and 324.

For clarity, we have divided Reviewer #1’s comments into numbered points.

Reviewer #1: This paper is of importance for women considering fertility.

Smoking and other addictive substances affect many patients. Warning of nicotine exposure is a continuous challenge. 

(1) The importance of segregating patients with PCO who have high AMH enable to better assess the non PCO group. However lack of definite information on hormonal contraceptives of any kind that can reduce AMH while in use weaken the paper. With BCP or implant a 25% decrease was noted. This has to be examined before conclusion of the results are made. Possibly this is a major contributor for borderline results obtained. 

Response: Thank you for your comments. As acknowledged in the manuscript, data on prior use of hormonal contraceptive was not available in these women, and thus the impact on effect estimate is unknown. However, if some of these women were on hormonal contraceptives prior to seeking fertility treatment, we expect the effect to be evenly distributed among smokers and nonsmokers and therefore bias the effect size towards the null. 

Use of oral contraceptives, especially long-term use, has been associated with moderate and reversible decrease in measures of ovarian reserve [1-6]. Landersoe et.al (2020) in a prospective cohort study on long term use of combine oral contraceptive (COC) with mean duration of use of 8.0 years, demonstrated a significant increase in measures of ovarian reserve after discontinuation. The AMH level increased by 53% (95% confidence interval [CI] 1.40–1.68, P < 0.001); AFC by 41% (95% CI 1.30–1.52, P < 0.001); and ovarian volume increased from 2.4 to 5.8 ml (P < 0.001) within 2 months after discontinuation of COC [1]. Similarly, Letourneau et.al (2017), in a longitudinal study of 743 women undergoing fertility preservation demonstrated that a break (average of 4 months) in combined hormonal contraception (CHC) use prior to ovarian stimulation resulted in increased oocyte yield with approximately twice as many oocytes per initial AFC as with no break (2.8 ± 3.8 vs. 1.4 ± 0.9, P < 0.001).

According to a paper published by Landersoe et.al in 2020 [6] , ovarian reserve testing can be considered accurate 2 months after discontinuation of COC. 

In the LOUSSI study, a subset of the women used in this analysis were followed prospectively for conception and pregnancy outcomes (N=184); it therefore follows that these women were not currently using any form of hormonal contraceptive. The estimates of association among this subset were similar to the original results (See Table in Response to Reviewer letter), though confidence intervals were slightly wider.

(2) Also AMH level that is consistent with PCO needs to be better defined in the methods section based on currently accepted criteria. 

Response: The PCOS status as noted in the manuscript was extracted from the medical record, documented as a current diagnosis or as a previous diagnosis self-reported by the patient. AMH levels are known to be higher, on average, among PCOS patients, and that was also demonstrated in our study (5.35±4.36 vs. 2.46±2.94 ng/mL) (mean±SD).

(3) Showing the results in a graph format. Where individual results showed provide a better description of data.

Response: We are not sure what the reviewer is suggesting. We have separated Table 2 into Tables 2 and 3 for clarity.

Reviewer #2: This is a very interesting article based on the results of a cross sectional study conducted with 207 women enrolled in the LOUSSI Study (Louisville Tobacco Smoke Exposure Genetic Susceptibility and Infertility) that fulfils all the inclusion criteria to be accepted for review in this Journal

I believe that the study is important, however I would like the authors to clarify some points:

1) The first sentence of the Results in the Abstract section (lines 28-29 Page 2) should be more clear because seems contradictory with the next one. Maybe authors could mention "Current smoking status either passive or active measured by a urinary cotinine assay..."

Response: The first sentence of the Results in the Abstract section (lines 28-29 Page 2) has been updated to clarify that the current smoking status assessment was based on urinary cotinine assay.

2) Although the previous history of pelvic surgery and of endometriosis has been considered the study has the drawback that has been conducted in a population seeking infertility counselling or treatment meaning that are women with infertility problems some of them could be related to diminished ovarian reserve not related to tobacco use. I wonder if authors have information of the same studies conducted in general population without fertility problems.

Response: Some of the studies referenced in the manuscript were conducted in a general population with sample sizes ranging between 913 to 3773 [7-12], but results were inconsistent.

3) It is very interesting the NAT2 acetylation status screening performed and it is a pity that due to the reduced sample size no differences have been observed in relation to the slow, intermediate or rapid categories

Response: Thank you; this limitation has been acknowledged in the manuscript.

4) I am surprised that 37.9 % of never smokers presented cotinine levels > 14 ng/ml. I would like authors to further comment on this finding. Maybe they should be more critical on the validity of this cotinine assay because it proved unable to discriminate what the clinical questionnaire on tobacco exposure did.

Response: Thank you for your observation. While the table header was labeled as row%, the percentages calculated were actually column percentages. This error has been fixed and the table clearly depicts the row % for cotinine concentration in urine by self-reported smoking status. The percentage of never smokers who presented cotinine levels > 14 ng/ml is 15%. This could be due to secondhand exposure or dishonest reporting.

5) Tables are busy and perhaps some of them can be simplified.

Response: To simplify the tables, the multivariable modeling results for the association of smoking with diminished ovarian result has been split into two tables: Table 2 for the odds ratios and Table 3 for AMH levels. The table for the NAT2 Interaction Analysis was updated to Table 4.

References

1. Landersoe SK, Birch Petersen K, Sørensen AL, Larsen EC, Martinussen T, Lunding SA et al. Ovarian reserve markers after discontinuing long-term use of combined oral contraceptives. Reproductive BioMedicine Online. 2020;40(1):176-86. doi:https://doi.org/10.1016/j.rbmo.2019.10.004.

2. Letourneau JM, Cakmak H, Quinn M, Sinha N, M IC, Rosen MP. Long-term hormonal contraceptive use is associated with a reversible suppression of antral follicle count and a break from hormonal contraception may improve oocyte yield. Journal of assisted reproduction and genetics. 2017;34(9):1137-44. doi:10.1007/s10815-017-0981-8.

3. Hariton E, Shirazi TN, Douglas NC, Hershlag A, Briggs SF. Anti-Mullerian hormone levels among contraceptive users: evidence from a cross-sectional cohort of 27,125 individuals. Am J Obstet Gynecol. 2021;225(5):515 e1- e10. doi:10.1016/j.ajog.2021.06.052.

4. Birch Petersen K, Hvidman HW, Forman JL, Pinborg A, Larsen EC, Macklon KT et al. Ovarian reserve assessment in users of oral contraception seeking fertility advice on their reproductive lifespan. Hum Reprod. 2015;30(10):2364-75. doi:10.1093/humrep/dev197.

5. Bentzen JG, Forman JL, Pinborg A, Lidegaard Ø, Larsen EC, Friis-Hansen L et al. Ovarian reserve parameters: a comparison between users and non-users of hormonal contraception. Reproductive BioMedicine Online. 2012;25(6):612-9. doi:https://doi.org/10.1016/j.rbmo.2012.09.001.

6. Landersoe SK, Forman JL, Birch Petersen K, Larsen EC, Nøhr B, Hvidman HW et al. Ovarian reserve markers in women using various hormonal contraceptives. The European Journal of Contraception & Reproductive Health Care. 2020;25(1):65-71. doi:10.1080/13625187.2019.1702158.

7. White AJ, Sandler DP, D'Aloisio AA, Stanczyk F, Whitworth KW, Baird DD et al. Antimullerian hormone in relation to tobacco and marijuana use and sources of indoor heating/cooking. Fertil Steril. 2016;106(3):723-30. doi:10.1016/j.fertnstert.2016.05.015.

8. Plante BJ, Cooper GS, Baird DD, Steiner AZ. The impact of smoking on antimullerian hormone levels in women aged 38 to 50 years. Menopause (New York, NY). 2010;17(3):571-6. doi:10.1097/gme.0b013e3181c7deba.

9. Dolleman M, Verschuren WM, Eijkemans MJ, Dolle ME, Jansen EH, Broekmans FJ et al. Reproductive and lifestyle determinants of anti-Mullerian hormone in a large population-based study. The Journal of clinical endocrinology and metabolism. 2013;98(5):2106-15. doi:10.1210/jc.2012-3995.

10. Schuh-Huerta SM, Johnson NA, Rosen MP, Sternfeld B, Cedars MI, Reijo Pera RA. Genetic variants and environmental factors associated with hormonal markers of ovarian reserve in Caucasian and African American women. Hum Reprod. 2012;27(2):594-608. doi:10.1093/humrep/der391.

11. Hawkins Bressler L, Bernardi LA, De Chavez PJ, Baird DD, Carnethon MR, Marsh EE. Alcohol, cigarette smoking, and ovarian reserve in reproductive-age African-American women. Am J Obstet Gynecol. 2016;215(6):758 e1- e9. doi:10.1016/j.ajog.2016.07.012.

12. Radin RG, Hatch EE, Rothman KJ, Mikkelsen EM, Sorensen HT, Riis AH et al. Active and passive smoking and fecundability in Danish pregnancy planners. Fertil Steril. 2014;102(1):183-91.e2. doi:10.1016/j.fertnstert.2014.03.018.

---

## [Decision Letter · Decision Letter 1]

6 Oct 2022

PONE-D-22-13340R1ASSOCIATION BETWEEN CIGARETTE SMOKING AND OVARIAN RESERVE AMONG WOMEN SEEKING FERTILITY CAREPLOS ONE

Dear Dr. Oladipupo,

Thank you for submitting your manuscript to PLOS ONE. After careful consideration, we feel that it has merit but does not fully meet PLOS ONE’s publication criteria as it currently stands. Therefore, we invite you to submit a revised version of the manuscript that addresses the points raised during the review process.Please submit your revised manuscript by Nov 20 2022 11:59PM. If you will need more time than this to complete your revisions, please reply to this message or contact the journal office at plosone@plos.org. Please include the following items when submitting your revised manuscript:A rebuttal letter that responds to each point raised by the academic editor and reviewer(s). You should upload this letter as a separate file labeled 'Response to Reviewers'.A marked-up copy of your manuscript that highlights changes made to the original version. You should upload this as a separate file labeled 'Revised Manuscript with Track Changes'.An unmarked version of your revised paper without tracked changes. You should upload this as a separate file labeled 'Manuscript'.If applicable, we recommend that you deposit your laboratory protocols in protocols.io to enhance the reproducibility of your results. Protocols.io assigns your protocol its own identifier (DOI) so that it can be cited independently in the future. For instructions see: https://journals.plos.org/plosone/s/submission-guidelines#loc-laboratory-protocols. Additionally, PLOS ONE offers an option for publishing peer-reviewed Lab Protocol articles, which describe protocols hosted on protocols.io. Read more information on sharing protocols at https://plos.org/protocols?utm_medium=editorial-email&utm_source=authorletters&utm_campaign=protocols.

We look forward to receiving your revised manuscript.

Kind regards,

Jing Zhang

Academic Editor

PLOS ONE

Journal Requirements:

Reviewers' comments:

Reviewer's Responses to Questions

**Comments to the Author**

1. If the authors have adequately addressed your comments raised in a previous round of review and you feel that this manuscript is now acceptable for publication, you may indicate that here to bypass the “Comments to the Author” section, enter your conflict of interest statement in the “Confidential to Editor” section, and submit your "Accept" recommendation.

Reviewer #2: (No Response)

Reviewer #3: All comments have been addressed

2. Is the manuscript technically sound, and do the data support the conclusions?

Reviewer #2: Yes

Reviewer #3: Yes

3. Has the statistical analysis been performed appropriately and rigorously? 

Reviewer #2: Yes

Reviewer #3: Yes

4. Have the authors made all data underlying the findings in their manuscript fully available?

Reviewer #2: Yes

Reviewer #3: Yes

5. Is the manuscript presented in an intelligible fashion and written in standard English?

Reviewer #2: Yes

Reviewer #3: Yes

6. Review Comments to the Author

Reviewer #2: I have carefully read the new version of this article. Authors have included the suggested changes and explanations. I believe that in its present form this article is suitable for publication.

Reviewer #3: Dear Authors,

I have read and mostly appreciated your article titled Association between cigarette smoking and ovarian reserve among women seeking fertility care, in which you have laid out a somewhat thorough analysis as to the linkage between heavy smoking and poor ovarian reserve.

I believe the manuscript has strengths: it is competently structured and coherently enunciates meaningful findings. The methodology is sound, as far as I was able to determine, and leads to insightful deductions and cogently delineated conclusions; relevance, thoroughness and well-crafted structure are certainly pros with the article as well.

The weakness I should point to is the scarce reference pool and outdated sources.

Among the studies laying out the AMH-ovarianreserve-smoking correlation, which is still based on rather inconclusive data, the authors have left out several recent ones that need to be drawn upon and could contribute to better contextualization and broader scope overall.

Consider, for instance:

Bhide P, Timlick E, Kulkarni A, Gudi A, Shah A, Homburg R, Acharya G. Effect of cigarette smoking on serum anti-Mullerian hormone and antral follicle count in women seeking fertility treatment: a prospective cross-sectional study. BMJ Open. 2022 Mar 31;12(3):e049646. doi: 10.1136/bmjopen-2021-049646.

Sansone M, Zaami S, Cetta L, Costanzi F, Signore F. Ovotoxicity of smoking and impact on AMH levels: a pilot study. Eur Rev Med Pharmacol Sci. 2021 Aug;25(16):5255-5260. doi: 10.26355/eurrev_202108_26545.

I think that the article is a commendable contribution to a highly relevant research field and ought to be greenlighted for publication with minor revisions.

Sincerely.

7. PLOS authors have the option to publish the peer review history of their article (what does this mean?). If published, this will include your full peer review and any attached files.

Reviewer #2: **Yes: **Pedro N. Barri

Reviewer #3: No

---

## [Author Response · Author response to Decision Letter 1]

12 Nov 2022

Response to Reviewers.

Journal Editor- Journal Requirements:

Response: The reference list has been reviewed and confirmed to be complete and correct. None of the papers cited in the manuscript have been retracted. Addition of three new relevant publications to the reference list was acknowledged in the rebuttal letter and revised cover letter accompanying the revised manuscript.

A paragraph referring to Table 3 was added to the manuscript and the table was moved below the paragraph to comply with the PLOS ONE submission guidelines.

Reviewer #2: I have carefully read the new version of this article. Authors have included the suggested changes and explanations. I believe that in its present form this article is suitable for publication.

Response: Thank you for your comments.

Reviewer #3: Dear Authors,

I have read and mostly appreciated your article titled Association between cigarette smoking and ovarian reserve among women seeking fertility care, in which you have laid out a somewhat thorough analysis as to the linkage between heavy smoking and poor ovarian reserve.

I believe the manuscript has strengths: it is competently structured and coherently enunciates meaningful findings. The methodology is sound, as far as I was able to determine, and leads to insightful deductions and cogently delineated conclusions; relevance, thoroughness and well-crafted structure are certainly pros with the article as well.

The weakness I should point to is the scarce reference pool and outdated sources.

Among the studies laying out the AMH-ovarianreserve-smoking correlation, which is still based on rather inconclusive data, the authors have left out several recent ones that need to be drawn upon and could contribute to better contextualization and broader scope overall.

Consider, for instance:

Bhide P, Timlick E, Kulkarni A, Gudi A, Shah A, Homburg R, Acharya G. Effect of cigarette smoking on serum anti-Mullerian hormone and antral follicle count in women seeking fertility treatment: a prospective cross-sectional study. BMJ Open. 2022 Mar 31;12(3):e049646. doi: 10.1136/bmjopen-2021-049646.

Sansone M, Zaami S, Cetta L, Costanzi F, Signore F. Ovotoxicity of smoking and impact on AMH levels: a pilot study. Eur Rev Med Pharmacol Sci. 2021 Aug;25(16):5255-5260. doi: 10.26355/eurrev_202108_26545.

I think that the article is a commendable contribution to a highly relevant research field and ought to be greenlighted for publication with minor revisions.

Sincerely.

Response: Thank you for your comments. The reference pool has been reviewed and updated with 3 new relevant publications on the association of smoking with ovarian reserve. These recent publications do not change the overall conclusions of the manuscript.

New References

53. Bhide P, Timlick E, Kulkarni A, Gudi A, Shah A, Homburg R et al. Effect of cigarette smoking on serum anti-Mullerian hormone and antral follicle count in women seeking fertility treatment: a prospective cross-sectional study. BMJ open. 2022;12(3):e049646. doi:10.1136/bmjopen-2021-049646.

54. Sansone M, Zaami S, Cetta L, Costanzi F, Signore F. Ovotoxicity of smoking and impact on AMH levels: a pilot study. European review for medical and pharmacological sciences. 2021;25(16):5255-60. doi:10.26355/eurrev_202108_26545.

55. Mitchell JM, Fee N, Roopnarinesingh R, Mocanu EV. Investigating the relationship between body composition, lifestyle factors, and anti-Müllerian hormone serum levels in women undergoing infertility assessment. Irish journal of medical science. 2022. doi:10.1007/s11845-022-03148-x.

---

## [Decision Letter · Decision Letter 2]

29 Nov 2022

ASSOCIATION BETWEEN CIGARETTE SMOKING AND OVARIAN RESERVE AMONG WOMEN SEEKING FERTILITY CARE

PONE-D-22-13340R2

Dear Dr. Oladipupo,

We’re pleased to inform you that your manuscript has been judged scientifically suitable for publication and will be formally accepted for publication once it meets all outstanding technical requirements.

Kind regards,

Jing Zhang

Academic Editor

PLOS ONE

**Comments to the Author**

1. If the authors have adequately addressed your comments raised in a previous round of review and you feel that this manuscript is now acceptable for publication, you may indicate that here to bypass the “Comments to the Author” section, enter your conflict of interest statement in the “Confidential to Editor” section, and submit your "Accept" recommendation.

Reviewer #3: All comments have been addressed

2. Is the manuscript technically sound, and do the data support the conclusions?

Reviewer #3: Yes

3. Has the statistical analysis been performed appropriately and rigorously? 

Reviewer #3: Yes

4. Have the authors made all data underlying the findings in their manuscript fully available?

Reviewer #3: Yes

5. Is the manuscript presented in an intelligible fashion and written in standard English?

Reviewer #3: Yes

6. Review Comments to the Author

Reviewer #3: Dear Authors,

I believe that you have rather successfully improved the manuscript and made it more comprehensive and well-rounded overall.

In light of the article's undeniable strengths and research value in a very relevant research field, I am going to recommend approval for publication.

Best regards.

7. PLOS authors have the option to publish the peer review history of their article (what does this mean?). If published, this will include your full peer review and any attached files.

Reviewer #3: No

---

## [Editor Report · Acceptance letter]

4 Dec 2022

PONE-D-22-13340R2 

Association between cigarette smoking and ovarian reserve among women seeking fertility care 

Dear Dr. Oladipupo:

I'm pleased to inform you that your manuscript has been deemed suitable for publication in PLOS ONE. Congratulations! Your manuscript is now with our production department. 

Kind regards, 

on behalf of

Dr. Jing Zhang 

Academic Editor

PLOS ONE